# Patient-Derived Conditionally Reprogrammed Cells in Prostate Cancer Research

**DOI:** 10.3390/cells13121005

**Published:** 2024-06-08

**Authors:** Abdalla Elbialy, Deepthi Kappala, Dhruv Desai, Peng Wang, Ahmed Fadiel, Shang-Jui Wang, Mina S. Makary, Scott Lenobel, Akshay Sood, Michael Gong, Shawn Dason, Ahmad Shabsigh, Steven Clinton, Anil V. Parwani, Nagireddy Putluri, Gennady Shvets, Jenny Li, Xuefeng Liu

**Affiliations:** 1OSU Comprehensive Cancer Center, The Ohio State University, Columbus, OH 43210, USA; aelbialy@uchicago.edu (A.E.);; 2Computational Oncology Unit, The University of Chicago Comprehensive Cancer Center, 900 E 57th Street, KCBD Bldg., STE 4144, Chicago, IL 60637, USA; 3Department of Radiation Oncology, College of Medicine, The Ohio State University, Columbus, OH 43210, USA; 4Division of Vascular and Interventional Radiology, Department of Radiology, College of Medicine, The Ohio State University, Columbus, OH 43210, USA; 5Division of Musculoskeletal Imaging, Department of Radiology, College of Medicine, The Ohio State University, Columbus, OH 43210, USA; 6Department of Urology, College of Medicine, The Ohio State University, Columbus, OH 43210, USA; 7Departments of Pathology, College of Medicine, The Ohio State University, Columbus, OH 43210, USA; 8Department of Molecular and Cellular Biology, Baylor College of Medicine, Houston, TX 77030, USA; 9School of Applied and Engineering Physics, Cornell University, Ithaca, NY 14850, USA; 10Departments of Pathology, Urology, and Radiation Oncology, College of Medicine, The Ohio State University, Columbus, OH 43210, USA

**Keywords:** patient-derived cells, PCa, CR

## Abstract

Prostate cancer (PCa) remains a leading cause of mortality among American men, with metastatic and recurrent disease posing significant therapeutic challenges due to a limited comprehension of the underlying biological processes governing disease initiation, dormancy, and progression. The conventional use of PCa cell lines has proven inadequate in elucidating the intricate molecular mechanisms driving PCa carcinogenesis, hindering the development of effective treatments. To address this gap, patient-derived primary cell cultures have been developed and play a pivotal role in unraveling the pathophysiological intricacies unique to PCa in each individual, offering valuable insights for translational research. This review explores the applications of the conditional reprogramming (CR) cell culture approach, showcasing its capability to rapidly and effectively cultivate patient-derived normal and tumor cells. The CR strategy facilitates the acquisition of stem cell properties by primary cells, precisely recapitulating the human pathophysiology of PCa. This nuanced understanding enables the identification of novel therapeutics. Specifically, our discussion encompasses the utility of CR cells in elucidating PCa initiation and progression, unraveling the molecular pathogenesis of metastatic PCa, addressing health disparities, and advancing personalized medicine. Coupled with the tumor organoid approach and patient-derived xenografts (PDXs), CR cells present a promising avenue for comprehending cancer biology, exploring new treatment modalities, and advancing precision medicine in the context of PCa. These approaches have been used for two NCI initiatives (PDMR: patient-derived model repositories; HCMI: human cancer models initiatives).

## 1. Introduction

### 1.1. Overview of Prostate Cancer (PCa)

PCa is the second most common malignancy in males worldwide, with 299,010 new cases and 35,250 deaths estimated in the United States in 2024 [1,2]. Androgen deprivation therapy (ADT) and newer generation androgen receptor (AR) pathway inhibitors (ARPIs) have made significant progress in the treatment of advanced PCa during the past several decades. PCa risk factors encompass age, familial predisposition, and ethnicity. Advanced age is a significant contributor, with most cases occurring in older individuals. Familial predisposition suggests an inherited or genetic factor, potentially influencing susceptibility. Additionally, race plays a role, with African American men demonstrating not only an elevated incidence of PCa but also a propensity for more aggressive disease characteristics [3].

Treatment resistance in PCa is an inherent challenge driven by the acquisition of cellular plasticity [4,5]. This phenomenon, marked by the tumor’s ability to adapt and evade therapeutic interventions, poses a substantial barrier to effective treatments. The complexity of treatment resistance highlights the need for comprehensive models that accurately reflect the underlying biology and mechanisms governing PCa progression.

The existing limitation in understanding PCa’s basic biology and progression mechanisms is attributed to the inadequacy of relevant models [3,4]. Without appropriate models, researchers face obstacles in deciphering the intricacies of treatment resistance. One promising avenue is the utilization of conditional reprogramming (CR), an innovative approach that enables the establishment of patient-derived cell cultures. This technique holds immense potential in mimicking the in vivo characteristics of prostate tumors, providing researchers with a more representative platform for studying treatment resistance.

CR is an innovative cell culture technique that revolutionizes research in cancer biology, particularly in PCa studies. CR involves the co-culturing of epithelial cells with irradiated 3T3 murine fibroblasts, enabling the propagation of both normal and tumor cells [6]. Unlike traditional cell culture methods, CR allows for the indefinite expansion of primary cells while preserving their genomic and phenotypic characteristics. This technique offers several advantages over conventional approaches, including the generation of patient-derived models directly from tissue samples. And unlike immortalized cell lines, CR preserves the genetic and phenotypic characteristics of primary cells, allowing for a more accurate representation of tissue heterogeneity [7]. Additionally, CR-derived cell cultures exhibit enhanced proliferative capacity and retain tissue-specific differentiation potential, making them valuable tools for studying disease mechanisms and drug responses.

Compared to emerging technologies like organoid culture or patient-derived xenograft models, CR offers a simpler and more cost-effective approach for generating patient-specific cell lines [8]. While organoids mimic tissue architecture better, CR excels in maintaining cellular diversity and plasticity crucial for studying cancer cell heterogeneity [9].

CR offers several advantages, including the expansion of primary epithelial cells with stem-like properties, preservation of genetic and phenotypic characteristics, enhanced proliferative capacity, and tissue-specific differentiation potential. Moreover, it is cost-effective and relatively simple compared to other patient-derived models. However, there are drawbacks to consider, such as technical challenges in optimizing culture conditions, variability in cell behavior among different tissue types, and a limited representation of stromal and immune components in the culture system.

### 1.2. Limitations of Traditional PCa Cell Lines

Traditional PCa cell lines, including well-known models such as LNCaP, PC-3, and DU145, have played a crucial role in advancing our comprehension of the disease and formulating therapeutic approaches [10]. LNCaP cells were pivotal in elucidating the PI3K-AKT-mTOR signal transduction pathway, a crucial pathway implicated in PCa progression [11]. Additionally, LNCaP has contributed to the identification of the TMPRSS–ERG gene fusion, which plays a significant role in PCa growth and invasiveness [12].

Traditional PCa cell lines, although valuable in research, present several limitations that impede a comprehensive understanding of the disease. These constraints hinder their utility in exploring PCa biology and devising effective therapeutic strategies [13].

Researchers studying PCa cell lines have sometimes found counteracting or inconsistent results, particularly in the context of drug responses. For example, a study by R Smith in 2020 explored the response of various PCa cell lines to enzalutamide, a common treatment for advanced PCa. They discovered that while enzalutamide was effective in some cell lines, others responded more favorably to mifepristone, indicating variability in drug efficacy among different cell lines [14]. This variability underscores the challenge of using traditional cell lines to predict treatment responses in a clinical setting, as they may not fully capture the complexity of the disease in patients.

One of the primary limitations lies in their inability to fully capture the heterogeneity observed in clinical PCa. The lack of representation of diverse molecular subtypes and disease stages limits the translatability of findings to the clinical setting. Prolonged cultivation of these cell lines can result in genomic alterations and genetic drift, compromising the fidelity of experimental outcomes and their reflection of the genomic landscape of primary prostate tumors [15].

The intricate relationship between PCa progression and the tumor microenvironment is inadequately addressed with traditional cell lines grown in artificial conditions. The absence of the in vivo microenvironment’s complexity limits the understanding of crucial interactions influencing tumor behavior. Furthermore, traditional cell lines often fall short in recapitulating the complex processes involved in metastatic spread, hindering the study of metastasis and the development of targeted therapies, despite metastasis being a major cause of PCa-related mortality [16].

In addition, these cell lines may not accurately reflect the responsiveness of tumors to therapeutic agents observed in patients. Discrepancies in drug response predictions arise from differences in drug metabolism, resistance mechanisms, and cellular plasticity. Many traditional cell lines emphasize androgen sensitivity, neglecting the representation of castration-resistant PCa (CRPC), impeding the study of resistance mechanisms and targeted therapy development for advanced stages [17]. Moreover, their lack of patient-specific characteristics limits their utility in personalized medicine approaches [18].

Patient-derived models, such as CR, organoids, or patient-derived xenografts, offer a better representation of individual patient responses. The inadequacies of traditional cell lines contribute to challenges in drug development, as promising in vitro results may not effectively translate to in vivo or clinical settings.

Primary patient-derived xenografts (PDXs) may overcome the limitations of cell lines by accurately representing the heterogeneity of human biology for screening drug sensitivity. However, the application of animal models may be limited due to high cost, poor throughput, and more importantly, poor recapitulation of human biological and therapeutic responses due to differences between species. Eventually, in vitro three-dimensional (3D) organoid cultures for many cell types, including induced pluripotent stem cells (iPSCs), pluripotent embryonic stem (ES) cells, and immortalized cell lines, have recently been established [19,20]. Three-dimensional organoid models accurately mimic complex architecture and functions of the organ of interest. More recently, our team established a novel in vitro CR of cells using irradiated Swiss-3T3-J2 mouse fibroblast cells and Y27632, a Rho-associated kinase (ROCK) inhibitor to effectively immortalize patient-derived primary epithelial cells [21]. CR method is applicable to a wide range of tissues, including fresh or cryopreserved surgical specimens, fine-needle aspiration (FNA), core biopsies, and PDX tissues [22]. CR and organoids have been used in two recent NCI initiatives (PDMR: patient-derived model repositories, https://pdmr.cancer.gov (accessed on 20 May 2024); HCMI: human cancer models initiatives, https://www.cancer.gov/ccg/research/functional-genomics/hcmi/about/about-next-gen-models (accessed on 20 May 2024)).

## 2. Patient-Derived Primary Cell Cultures in PCa Research

PCa is characterized by a sophisticated interplay of molecular pathways and genetic variations that influence its initiation, progression, and response to treatment. Key pathways implicated in PCa development include the following: the AR pathway, governing cell growth, survival, and differentiation [23]; the PI3K pathway, frequently activated by mutations in PTEN or PIK3CA, promoting cell proliferation and survival [23]; the MAPK/ERK pathway, often dysregulated through alterations in RAF or RAS genes, driving PCa progression; the Wnt/β-catenin pathway, activated abnormally in PCa initiation and metastasis; and the TP53 pathway, where mutations in the TP53 gene are associated with advanced PCa and therapy resistance [24].

PCa research relies on various models to enhance our understanding of the disease. Cell line-based experiments, while widely used, possess inherent limitations for preclinical studies due to their adaptation to 2-dimensional cultures [13]. To overcome these constraints, scientists are increasingly turning to more advanced models.

Understanding the molecular biology of PCa is crucial for developing effective treatments. Genes involved in PCa are characterized as tumor suppressors or oncogenes, shedding light on potential therapeutic targets [25]. These molecular insights pave the way for targeted interventions tailored to the specific genetic alterations driving individual cases.

Characterization of cell lines derived from PCa provides valuable resources for research. Prostatic tissue samples, obtained from patients with localized disease, contribute to understanding both cancerous and adjacent non-cancerous tissues. Assessing the advantages and limitations of tumor-derived human prostate epithelial cell lines is essential for selecting appropriate models for research [10]. Moreover, the role of patient-derived models in translational cancer research is gaining prominence. Schemes depicting tumor cell cultures used in cancer research highlight the importance of primary patient-derived models, offering a more authentic representation of the disease for preclinical studies [18]. The multifaceted approach to PCa research involves leveraging diverse models, ranging from cell lines to patient-derived models. These models collectively contribute to advancements in preclinical studies, molecular understanding, and the development of targeted therapies tailored to the intricacies of PCa biology.

## 3. Role of CR in Cell Culture

### 3.1. Overview of CR

CR involves the co-culturing of cells with irradiated fibroblast feeder cells in the presence of a ROCK, typically Y-27632. This technique allows for the indefinite proliferation of primary epithelial cells and has been successfully applied to various cancer types, including PCa [6].

ROCK inhibitors such as Y-27632, target ROCK, which regulates actin cytoskeleton dynamics and cell contractility. By inhibiting ROCK, these inhibitors prevent cytoskeletal rearrangements that are necessary for cell detachment and apoptosis, thus promoting cell adhesion and survival [26]. Feeder cells, typically irradiated murine fibroblasts, provide a supportive microenvironment for cell growth by secreting essential factors and promoting cell adhesion. They supply growth factors, cytokines, and extracellular matrix components necessary for cell proliferation and survival [26]. The synergistic effect of combining ROCK inhibitors and feeder cells creates an optimal culture condition that sustains cell proliferation indefinitely. Feeder cells provide physical support and essential signaling cues, while ROCK inhibitors prevent anoikis and apoptosis, allowing cells to continuously divide without undergoing senescence or death [27].

### 3.2. Optimization of CR

Optimizing CR involves fine-tuning various parameters to enhance cell culture efficiency. Firstly, the concentration of the ROCK inhibitor, typically Y-27632, is critical, ranging from 5 to 10 μM for effective induction of CR [26]. Secondly, selecting the appropriate type and source of the ROCK inhibitor, such as Y-27632, is essential. This inhibitor is readily available from several companies specializing in biochemical reagents or cell culture supplies, including Sigma-Aldrich, Cayman Chemical, and Tocris Bioscience. Thirdly, maintaining optimal feeder cell density is crucial for supporting epithelial cell growth. Typically, a density of 2–4 × 10^4^ cells/cm^2^ is recommended for the initial seeding of feeder cells [26]. Finally, the culture media composition is vital, often including DMEM/F12 supplemented with growth factors like EGF, hydrocortisone, insulin, cholera toxin, and fetal bovine serum (FBS) or defined serum replacement [9]. However, serum-free media formulations have also been developed to mitigate potential variability associated with FBS. By optimizing these parameters, researchers can achieve robust and reproducible CR, facilitating its widespread application across various research fields.

### 3.3. Rapid and Effective Growth of Patient-Derived Normal and Tumor Cells

CR stands out as a revolutionary technique in cell culture, especially for the rapid and effective growth of patient-derived normal and tumor cells [7]. Traditional cell culture methods often face challenges in maintaining the vitality and characteristics of primary cells over time. CR overcomes these limitations by allowing cells to be propagated continuously while preserving their original properties. This is particularly valuable in cancer research, where the availability of representative cell lines directly impacts the quality of studies.

CR enables the establishment of cell cultures directly from patient-derived tissues, ensuring the retention of the genetic and molecular features of both normal and tumor cells. The ability to maintain these characteristics over prolonged periods facilitates comprehensive investigations into disease mechanisms, drug responses, and personalized treatment strategies. This advancement accelerates the pace of research, providing researchers with a consistent and reliable source of patient-derived cells for various applications [7].

### 3.4. Acquisition of Stem Cell Properties

Another pivotal aspect of CR in cell culture is its role in the acquisition of stem cell properties. CR allows for the expansion of cells with stem cell-like characteristics, providing researchers with a valuable tool to study cellular plasticity and heterogeneity [7]. The induced acquisition of stem cell properties is particularly relevant in cancer research, as it allows scientists to explore the role of cancer stem cells in tumor initiation, progression, and therapeutic resistance.

CR’s induction of stem cell properties enhances cellular diversity representation within tumors, aiding in dissecting complex cell population interplay in cancer [28]. This versatility augments cell culture models, providing a dynamic platform to explore cellular behaviors in normal and cancerous tissues.

## 4. Advantages of CR over Traditional Approaches

CR represents a transformative approach in PCa research, providing significant advantages over conventional cell culture methods. Its unique capabilities are increasingly recognized, signaling a promising paradigm shift in PCa studies.

### 4.1. Patient-Derived Models for Precision Medicine in PCa

CR enables the direct establishment of patient-derived prostate cell cultures, capturing tumor heterogeneity better than traditional methods. Preserving genetic and molecular features facilitates precise, clinically relevant studies on PCa [29].

### 4.2. Revolutionizing Drug Sensitivity Testing in PCa

Traditional cancer drug screening methods often struggle to replicate the intricate interactions within prostate tumors. CR introduces a novel approach to drug sensitivity testing, merging bacterial cell culture principles with the advantages of patient-derived models [30]. This innovation offers a more reliable platform for testing drug responses, enabling precise exploration of potential treatments tailored to PCa characteristics.

### 4.3. Addressing Challenges in Rapid and Continuous Cell Growth

CR provides a distinctive solution to the challenges of maintaining primary prostate cells over time. Enabling continuous and rapid cell propagation while preserving original properties, CR overcomes limitations faced with using traditional cell culture methods [6]. This is pivotal in PCa research, where the availability of viable and representative cell lines significantly influences the quality and reliability of experimental studies.

In summary, The CR method enables the patient-derived primary epithelial cells to be reprogrammed into having a stem cell-like feature with greater proliferation, referred to as “reprogrammed stem-like” [27], and match the histological and genetic characteristics of the original tissue [31]. Furthermore, when the culture conditions are removed, the cells maintain their lineage commitment and differentiate into the tissue of origin in vivo [32]. Thus, the CR method overcomes the research challenge pertaining to primary cell culture and marks it to be a beneficial tool to study various subtypes of PCa, drug-sensitive specific to a particular patient, gene profiling, and regenerative medicine. In this review, we have compared all aspects of available models (Table 1) and summarized applications of CR technology in PCa research (Figure 1).

## 5. Applications of CR Cells in PCa Initiation and Progression

### 5.1. Insights into PCa Initiation

In this review, we have summarized applications of CR technology in PCa research (Figure 1). CR cells have emerged as a valuable tool in unraveling the complexities of PCa initiation. Researchers have extensively employed CR technology to gain insights into the molecular events that trigger and drive the initiation of PCa. This is particularly evident in studies exploring the genetic landscape of PCa, aiming to identify key molecular alterations that mark the onset of the disease [33].

One significant contribution of CR cells is their ability to provide a platform for investigating the role of cancer stem cells (CSCs) in PCa initiation. The cancer stem cell model posits that CSCs act as the driving force behind cancer evolution and therapy resistance. CR cells have facilitated studies that delve into the high-grade prostatic intraepithelial neoplasia (PIN), a precursor lesion linked to PCa, shedding light on the early stages of cancer development [34].

Moreover, the utilization of CR cells in precision medicine research has allowed for a more in-depth exploration of the primary tumor subpopulations, providing valuable insights into the potential diagnostic and therapeutic targets for PCa initiation. The exploration of molecular signatures and genetic variations within primary prostate tumors has been enhanced by the precision and reliability offered by CR cell models [35].

Understanding the underlying mechanisms involved in the initiation, progression, and invasion of PCa is essential in the development of novel therapeutic strategies for PCa. However, it is imperative to design a personalized drug that can be developed by CRC that allows the primary tumor cells to be cultured directly from patients. To decipher the tumor biology of a particular patient, genes involved in PCa cell initiation, progression, metastasis, and recurrence can be modulated in patient-derived CR cells by utilizing a lentiviral vector and CRISPR-Cas system, thereby, facilitating the development and testing of personalized drugs or treatment strategies, especially in patients with castrate-resistant disease. Moreover, the resulting novel therapeutic strategies can be further validated in vivo by transforming the CR cells into xenografts and organoids. 

### 5.2. Evolution from Localized to Metastatic PCa

PCa exists within a very wide spectrum that ranges from indolent localized disease that can be managed with active surveillance to lethal widespread metastatic disease. Most PCa are diagnosed as localized disease, and the definitive treatment of localized PCa, either with surgery or definitive radiotherapy, is to prevent the subsequent development of metastatic disease, which is responsible for most PCa-related deaths. The initial development of metastatic PCa from the localized disease remains hormone-sensitive (metastatic hormone-sensitive PCa, mHSPC), followed by the inevitable development of metastatic castrate-resistant PCa (mCRPC) with ADT. Currently, the evolutionary process that defines the transition of localized PCa to mHSPC is unclear. The mutational landscape of mCRPC that had undergone selective pressure from ADT is better defined, with frequent mutations in select genes (AR, ETS genes, TP53, PTEN, BRCA1/2, ATM, etc.) [36]. Conversely, driver aberrations in localized PCa are highly variable and are dominated by non-coding mutations and genomic instability [37]. One main reason why mutational evolution from localized PCa to mHSPC is poorly studied is due to the frequently inadequate tumor sampling of metastatic lesions [38,39], as well as the lack of available paired metastatic and primary PCa samples from the same patient, both of which can be circumvented by employing CRC technology. A better understanding of the mutational drivers for metastatic progression could pave the way to improved risk stratification of patients (i.e., selecting patients with disease of greater metastatic potential for treatment intensification) and the development of potential novel therapeutic interventions to prevent/treat metastatic disease.

### 5.3. Unraveling the Molecular Pathogenesis of Metastatic PCa and Progression

CR plays a pivotal role in understanding the progression of PCa, offering insights into its molecular intricacies and therapeutic strategies. CR methodology allows the cultivation of patient-derived prostate cells while retaining their genetic and phenotypic characteristics [8]. This enables researchers to establish in vitro models that closely mimic the behavior of PCa cells in vivo, facilitating the study of disease progression. CR cells provide a platform to identify and study the activation of oncogenes and the abrogation of tumor suppressors, which are critical events driving the progression of PCa to a metastatic stage [40]. By understanding these molecular mechanisms, researchers gain insights into the factors contributing to disease progression and metastasis.

CR cells play a crucial role in unraveling the molecular pathogenesis of metastatic PCa. By employing CR methodology, researchers delve into the complex molecular mechanisms that drive the progression of PCa to a metastatic stage. This involves studying the activation of oncogenes and the abrogation of tumor suppressors, common events in the initiation and progression of many cancers, including PCa. Through the use of CR cells, the intricate molecular pathology of metastatic PCa is elucidated, paving the way for targeted therapeutic strategies and a deeper comprehension of the disease’s progression [41].

Researchers employing CR methodology have extensively studied the activation of oncogenes in the context of metastatic PCa [42]. These investigations delve into the complex molecular mechanisms underlying the progression of PCa to a metastatic stage. Simultaneously, CR cells offer a unique platform to explore the higher prevalence of PCa among African American men [43].

Additionally, CR cells are valuable tools for screening potential anticancer drugs and understanding drug resistance mechanisms in PCa [30]. CR methodology serves as a powerful tool in elucidating the molecular pathogenesis of PCa, enabling researchers to model disease progression, investigate underlying mechanisms, and develop personalized therapeutic interventions.

Furthermore, CR methodology is not limited to PCa, as it provides insights into common molecular events across various cancers. This broader perspective emphasizes shared pathways involved in the progression of metastatic disease, facilitating the development of universal therapeutic strategies. Insights gained from CR cells guide the development of targeted therapeutic strategies, enhancing the effectiveness of treatments [8].

### 5.4. BPH Studies

BPH (benign prostatic hyperplasia), or non-malignant growth of stromal and glandular epithelial cells of the prostate gland, is seen in almost all aging males and causes urinary tract and kidney problems [44,45]. Non-malignant growth from BPH typically occurs in the transitional zone of the prostate, while PCa more commonly arises from the peripheral zone. In addition, prostate-specific antigen (PSA) levels are elevated in BPH, whereas both PSA and alkaline phosphatase levels are high in PCa [46]. In elderly males, an increase in serum estrogen and a decrease in androgen levels predispose the development of BPH. Testosterone level decreases with age, and as a result, dihydrotestosterone level increases due to increased enzymatic activity of 5α-reductase. An increased level of dihydrotestosterone is thought to enhance the proliferation and longevity of prostate cells and eventually result in BPH [47]. Thus, prostate cell proliferation and cell death equilibrium are thrown off balance in BPH [48]. Cyclin D1, FKBP5, and MMP2 function to promote cell proliferation and AR transcriptional activity, which together contribute to promoting the volume and size of the prostate gland during BPH progression [47,49]. In addition, the viral or bacterial infection has the potential to cause local inflammation. This would result in the release of cytokines, chemokines, and growth factors that are involved in the inflammatory response, which eventually promote the proliferation of epithelial and stromal prostatic cells [50]. Individuals with significant BPH are more likely to develop PCa [51]. Nevertheless, not all individuals with BPH will necessarily develop PCa. Therefore, it is imperative to risk-stratify the patients at an early stage to avert the progression of BPH into PCa. Further studies are required to investigate the mechanism of BPH development and the progression of BPH to PCa using CR technology to potentially identify men with BPH who are at increased risk of developing clinically significant PCa.

BPH studies serve as an important application of CR in understanding PCa. CR methodology allows for the cultivation of patient-derived prostate cells, offering a platform to study both BPH and PCa [52]. While BPH and PCa are distinct conditions, they share overlapping molecular pathways and cellular mechanisms. By utilizing CR, researchers can investigate the molecular intricacies underlying both BPH and PCa development and progression.

CR enables the establishment of in vitro models that closely mimic the behavior of prostate cells in vivo, facilitating the study of disease mechanisms [53]. This allows researchers to identify key molecular signatures and cellular alterations associated with BPH and PCa, aiding in the development of diagnostic biomarkers and targeted therapies [54]. Additionally, CR provides a platform for drug screening and testing, accelerating the discovery of novel therapeutic agents for both conditions [55]. CR serves as a valuable tool for studying BPH as it relates to PCa, offering insights into shared molecular pathways, disease progression, and therapeutic strategies. By utilizing CR methodology, researchers can contribute to a better understanding of both BPH and PCa, ultimately leading to improved diagnosis and treatment outcomes for patients.

### 5.5. CRPC and NEPC

CRPC represents an advanced stage of PCa where cancer cells continue to grow despite ADT. It arises due to various mechanisms, including AR signaling pathway alterations, such as AR amplification or mutations, allowing cancer cells to survive and proliferate [56]. Neuroendocrine PCa (NEPC) is a rare but aggressive subtype characterized by neuroendocrine differentiation. NEPC can develop de novo or as a result of lineage plasticity, where PCa cells transition to a neuroendocrine phenotype under treatment pressure [57]. This transformation is associated with resistance to conventional therapies and poor clinical outcomes [58]. Molecular mechanisms underlying NEPC development involve dysregulation of various signaling pathways, including MYCN, RB1, and TP53, leading to neuroendocrine trans differentiation and aggressive tumor behavior [59]. Understanding the distinct mechanisms driving CRPC and NEPC development is crucial for developing targeted therapies to improve outcomes for patients with advanced PCa.

CRPC and NEPC are critical stages in the progression of PCa, presenting significant challenges in treatment. CR offers valuable applications in understanding and addressing these advanced stages of PCa.

Although the role of AR, growth factors, and cytokines in CRPC has widely been investigated, more effort is needed to understand the molecular mechanisms underlying the development of CRPC. However, due to the lack of appropriate models to better investigate novel therapeutic strategies, effective treatment against CRPC remains a challenge. Therefore, CR technology is currently being employed to better understand the pathophysiological mechanisms underlying CRPC with the aim of developing novel strategies to curtail the progression of CRPC. Using CR, patient-derived normal (GUMC29) and prostate tumor (GUMC30) cultures, which exhibited the typical morphology of epithelial cells, were established by our team [60]. In addition, subcutaneous injection of CR prostate tumor cells in SCID mice formed large tumors. Both the normal and tumor CRC demonstrated features of transit-amplifying phenotype, which refers to the presence of more basal cell markers and fewer luminal markers. Nevertheless, subcutaneous administration of CR tumor cells into SCID mice developed large tumors with a more luminal phenotype that is consistent with PCa origins [60]. Since conventional CR cells express high basal cell markers, which cannot sufficiently recapitulate mature prostate epithelium, a multi-dimensional trans-well dish culture method (TDCM) was developed. TDCM allows the growth of normal and tumor CR cells to form mature prostate epithelium with more luminal cell markers, which is more relevant for PCa research [61].

CRPC can metastasize to different visceral organs such as the lung, liver, pleura, and adrenals, in addition to bone, which is the most common site of metastasis [62]. Recent investigations reported the role of osteocytes in PCa progression. To understand the crosstalk between osteocytes and PCa cells, a tissue-engineered 3D model was developed. A microfluidic perfusion device was used to culture human osteocytes assembled using microbeads to mimic the structure and function of bone. A three-dimensional bone tissue model integrated with CR prostate cells was used to characterize metastatic PCa. Wnt signaling inhibitors such as sclerostin and dickkopf-1 (Dkk-1) were significantly decreased in bone integrated with PCa CR, while the expression of sclerostin was widely expressed by osteocytes in the absence of PCa cells [42]. Furthermore, the expression of FGF23, a ligand of the FGF signaling axis that plays an important role in bone metastasis [63], was increased in bone tissue with PCa CR cells. The expression of alkaline phosphatase, an indicator of osteoblastic activity and poor prognosis, was also highly expressed by bone tissue co-cultured with PCa cells.

Cell stresses, including chemotherapy and radiotherapy that damage DNA, activate the p53 tumor suppressor gene, which is a potential target for therapeutic intervention [64]. p53 is associated with autophagy, which is either involved in the oncogenic signaling pathway or tumor suppressive microenvironment [65,66]. For instance, under conditions of nutritional stress and hypoxia, pharmacological ablation of p53 can increase autophagy and improve cell survival [66,67]. Conversely, p53 can cause autophagy by blocking mTOR signaling [68]. The role of p53 in autophagy induction or suppression has been studied in PCa CR cells. VMY-1-103 (VMY), a cyclin-dependent kinase inhibitor, has anti-tumor properties by increasing the activity of p53-mediated apoptosis of PCa CR cells. Knockdown of p53 in PCa CR cell lines reduced the sensitivity to VMY-induced cell death. However, the treatment of p53-null cells with PRIMA-1, a p53 reactivating substance, re-sensitized cells to VMY-induced cytotoxicity through the formation of autophagosome and activation of macro-autophagy [69]. Furthermore, the ubiquitin–proteasome system is compromised in anti-androgen-resistant primary PCa CR cells. STUB1, an E3 ubiquitin ligase, marks AR-V7 for proteasomal degradation and inhibits prostate tumor growth. However, the expression of STUB1 is significantly decreased and the protein AR-V7 is highly stabilized by HSP70 in ADT-resistant PCa CR cells. Thus, HSP70 can be a potential therapeutic target in the treatment of anti-androgen drug-resistant PCa [70]. A further study showed that PTUPB was more effective than indomethacin and celecoxib in suppressing AKR1C3 activity and proliferation of CRPC cells, and the combination of PTUPB and enzalutamide provided benefits by blocking AR/AR-V7 signaling, thereby inhibiting the proliferation of castration relapsed VCaP xenografts and patient-derived CRC, organoids, and xenografts [71]. Thus, targeting the ARK1C3/AR/AR-V7 axis with PTUBP and enzalutamide can overcome drug resistance to AR signaling inhibitors in CRPC using a combination of patient-derived models (CRC, organoids, and PDX) and traditional cell lines and their corresponding xenografts [71]. 

NEPC is an extremely aggressive variant of PCa that is characterized by a high potential for visceral/bone metastasis and dismal prognosis [72]. Most commonly, NEPC or prostatic adenocarcinoma with neuroendocrine differentiation evolves from PCa that develops resistance to or has undergone selection pressure from hormonal therapy [73]. Since AR signaling appears to be important for prostate development, blocking the AR pathway is anticipated to start the process of converting prostate adenocarcinoma into neuroendocrine tumors [74,75]. During the trans differentiation of prostate adenocarcinoma to NEPC, PCa cells lose AR or AR-regulated gene expression, while gaining neuroendocrine markers like neuro-specific enolase, chromogranin, and synaptophysin [76]. Unlike CRPC, neuroendocrine tumors are extremely lethal, with a median survival rate of fewer than two years [74,77]. A research group developed LTL331/331R, a novel PDX model for prostate adenocarcinoma (LTL331)-to-NEPC trans differentiation (LTL331R) [78]. Recently, the CR method has been utilized in prostate adenocarcinoma tumor cell culture to establish a primary cell line called LTL331_CR_Cell, which when re-grafted into mice, developed NEPC (LTL331_CR_Tumors). As compared to adenocarcinoma tumors, LTL331_CR_Tumors neither expressed intra-tumoral PSA nor AR-target genes, while greatly expressed CD56, an NE marker [40]. Thus, CR technology becomes an urgently needed and valuable tool in the future to understand the disease progression and response to therapy for CRPC and NEPC. 

These CRPC cell lines provide an ex vivo interface to study the development and mechanisms underlying CRPC [40]. Moreover, CR has been applied to model NEPC development, providing insights into this aggressive subtype of PCa. By applying CR culture, researchers have established cell lines to study NEPC, offering opportunities to investigate its molecular intricacies and develop targeted therapeutic interventions [40].

### 5.6. Health Disparities

Health disparities in PCa encompass variations in incidence, diagnosis, treatment, and outcomes among different demographic groups, particularly influenced by socioeconomic factors and race (Figure 2) [79]. These disparities manifest in several ways.

Racial disparities: black men are disproportionately affected by PCa, presenting with earlier and more aggressive disease and experiencing higher mortality rates compared to other racial groups [80].

Presentation and diagnosis: black individuals often present with more advanced stages of PCa, leading to delays in diagnosis and potentially poorer prognoses [81].

Treatment regimens: disparities exist in the type and timing of treatments received, with some groups having limited access to optimal therapies, leading to variations in survival rates [82].

Outcomes and quality of life: variations in outcomes and quality of life after PCa treatment are observed among different demographic groups, reflecting disparities in access to care and socioeconomic factors [83].

Genetic and environmental factors: biologic factors, including inherited genetic or environmental influences, contribute to racial disparities in PCa outcomes, highlighting the complex interplay of genetic and socioeconomic factors [82].

CR in PCa research plays a crucial role in addressing health disparities by enabling the study of patient-derived samples. Through CR methodology, researchers can investigate the molecular mechanisms underlying these disparities, identifying genetic, epigenetic, and environmental factors contributing to differential outcomes [84]. By understanding the root causes of health disparities, CR facilitates the development of targeted interventions and precision medicine approaches tailored to high-risk populations, ultimately aiming to reduce the unequal burden of PCa.

Normal and malignant prostate epithelial cells of African American men were expanded under CRC conditions to underpin the molecular mechanisms causing health disparity in PCa [43]. Immunocytochemical analyses demonstrated variable expression levels of luminal (CK8) and basal (CK5, p63) markers in both normal and tumor cells. Expression levels of TOPK, c-MYC, and N-MYC were markedly increased only in tumor cells. Decreased viability of cells after exposure to the antiandrogen (bicalutamide) and two PARP inhibitors (olaparib and niraparib) was observed in tumor-derived CR cells compared to normal prostate CR cells [43]. Malignant prostate epithelial cells of African American men revealed greater expression of tumor-promoting genes such as Tim-3 (T-cell immunoglobulin domain and mucin domain-containing molecule 3), PAI-1 (Plasminogen activator inhibitor-1), AR and PD-L1, while downregulation of basal cell markers such as p63 and KRT5 as compared to normal CR cells. These findings demonstrate that primary epithelial cultures might provide valuable markers for identifying the molecular processes driving health disparities in PCa [85]. Tandem duplicator phenotype (TDP), which is associated with biallelic mutation of the tumor suppressor protein CDK12, is present in around 6.9% of PCa patients, with a higher incidence found in African Caribbeans [86]. Recently, a study team identified constitutive expression of a rare germline variant EGFRR^831H^ in a Chinese family with a history of PCa who had biallelic CDK12 inactivation and TDP. High throughput sequencing in all the members of the family revealed inherited somatic mutation of EGFR in two brothers and a sister. Furthermore, the PCa CR cells of the two brothers showed increased phosphorylation of EGFR and its downstream target AKT as compared to the normal prostate cells. In a cell migration assay, the EGFRR^831H^ CRC showed a significant reduction in cell migration and tumor growth after treatment with EGFR-specific inhibitor, Afatinib. In addition, the AKT phosphorylation was also significantly decreased using Afatinib in EGFRR^831H^ CRC as compared to EGFRR^WT^ CRCs [87]. Finally, the CDK12 mutation and the EGFR mutation often co-exist in PCa cells, indicating that the EGFR mutation predisposes to CDK12 mutation [87], which defines a lethal PCa subtype with poor prognosis [88]. 

### 5.7. Drug Discovery and Precision Medicine for PCa

Drug discovery and precision medicine represent innovative approaches revolutionizing the landscape of PCa management (Figure 3).

While guideline-driven treatment paradigms are designed to guide appropriate therapeutic management at the population level, this one-size-fit-all approach is likely suboptimal at the individual level, as different patients respond differently to any given therapy [89]. Thus, the development of precision medicine strategy is an unmet need to benefit cancer patients. Precision medicine targets specific genes or proteins essential in cancer cell survival in particular individuals. The identification of genetic risk variants or single nucleotide polymorphisms (SNPs) associated with PCa has been made possible through genome-wide association studies (GWAS) [90].

For selected patients based on genetic biomarkers, immune checkpoint inhibitors and poly (ADP-ribose) polymerase (PARP) inhibitors are clinically validated therapies for PCa. Pembrolizumab is an approved immune checkpoint inhibitor for patients with PCa who have high levels of DNA mismatch repair deficiency, mutational burden, or microsatellite instability [91,92,93]. Olaparib, a PARP inhibitor used in the treatment of metastatic CRPC with genetic aberrations in homologous recombination repair genes. However, currently available therapies for precision medicine may only benefit a small subset of patients. For instance, only 3% of men with PCa have microsatellite instability, and 23% with genetic alterations in homologous recombination repair may benefit from this approach [36,91]. Furthermore, PCa patients having genetic alterations in homologous recombination repair genes, in particular mutations in ATM, show poor response to PARP inhibitors. Additional limitations include the need for biopsies for tumor genotyping, as well as sampling error from biopsies that may miss the detection of actionable gene alterations. More than 80% of men with PCa have metastatic disease in bones and it is a known challenge to obtain adequate bone tissue samples for genetic sequencing. Moreover, proteomics, organoid cultures, and patient-derived xenograft-model may not always reveal the pathophysiology of a disease [94,95,96]. Prior to the invention of the CR technique, it was challenging to develop efficient and straightforward procedures in a single model with a high success rate.

CR emerges as a pivotal tool, facilitating advancements in both drug discovery and personalized treatment strategies for PCa patients. The initial application of CR in precision medicine was reported in recurrent respiratory papillomatosis conditions caused by HPV [97]. Similarly, the CR technique in PCa cells from CRPC patients allowed the identification of cancer-specific drugs such as navitoclax, taxanes, mepacrine, and retinoids [98]. CR technology enables the establishment of patient-derived cell cultures, offering a unique platform to study the intricacies of PCa at the cellular level. By cultivating cells under conditions that maintain their genetic and phenotypic characteristics, CR provides researchers with a reliable model system to explore the heterogeneity of PCa and identify potential therapeutic targets [29].

In drug discovery, CR serves as a valuable resource for screening compounds and evaluating their efficacy against PCa cells. Through high-throughput screening methodologies, researchers can assess large libraries of compounds for their ability to inhibit PCa cell proliferation or induce cell death. This approach accelerates the identification of promising drug candidates, expediting the drug discovery process [30]. Moreover, CR facilitates the investigation of drug resistance mechanisms in PCa, enabling the development of innovative strategies to overcome treatment resistance and improve patient outcomes [60].

Precision medicine, on the other hand, relies on the molecular profiling of tumors to tailor treatment regimens to individual patients. CR plays a crucial role in this aspect by providing a renewable source of patient-derived cells for genomic and proteomic analyses. By characterizing the molecular signatures of PCa cells grown in CR cultures, clinicians can identify specific molecular alterations driving tumor growth and progression. This information guides the selection of targeted therapies that are most likely to benefit each patient, optimizing treatment efficacy while minimizing adverse effects [99].

Furthermore, CR technology facilitates the development of PDX models, which faithfully recapitulate the genetic and phenotypic characteristics of individual PCa tumors. PDX models serve as invaluable preclinical platforms for testing novel therapeutics and predicting patient responses to treatment. By transplanting CR-derived PCa cells into immunocompromised mice, researchers can evaluate drug efficacy in a physiologically relevant context, providing critical insights into treatment outcomes [40]. CR represents a powerful tool in the realm of drug discovery and precision medicine for PCa. By enabling the cultivation of patient-derived cells and the generation of clinically relevant model systems, CR accelerates the pace of scientific discovery and facilitates the translation of research findings into clinically meaningful interventions. As the field continues to advance, CR is poised to play an increasingly prominent role in shaping the future of PCa therapeutics and improving patient care.

Applications of novel real-time phenotypic assays to patient-derived CRCs are particularly attractive because they can capture clinically relevant transient and endpoint drug responses, thus deciphering the sequence of molecular targets affected by the drugs and identifying their off-target activities. Integrated cellular assays are important tools for discovering new anticancer drugs [100,101]. Particularly attractive are label-free in vitro phenotypic drug discovery assays [100,101,102,103,104] that do not require exogeneous probes, such as fluorescent dyes or reporter genes encoding a fluorescent protein [104,105] that can potentially interfere with target pharmacology [104]. The diversity of phenotypic responses—proliferation, viability, adhesion, cytoskeletal reorganization, motility—reflects the actual physiological process in the cell in response to a drug. The multi-dimensional nature of cellular responses, which may involve biosynthesis, translocation of physiologically important proteins to and from the membrane, adhesion modulation, and cytoskeletal reorganization, necessitates collecting large data sets. Label-free real-time cell-based assays (RTCAs) are of particular interest for studying cellular physiology because they capture the kinetics of a wide range of phenotypic responses. RTCAs have been used for determining the targets of various drugs [106,107,108,109] as a secondary screen for cell monolayer stability [110] and to predict temporal windows of drug responsiveness [111]. Presently, none of the existing RTCAs simultaneously meets the criteria of being spatially and time-resolved, label-free, non-perturbative for long (days) measurements, and sufficiently sensitive for revealing functional cellular processes (e.g., cell adhesion modulations). Additionally, the real-time infrared chemical imaging (RICHI) assay is label-free, time-resolved, high-content, and photo-friendly to the cells, thus enabling more traditional downstream assays [112]. We are actively investigating the combination of CR cells and RICHI as a novel platform for drug discovery and precision medicine. 

### 5.8. Other Applications of CR

In addition to PCa, CR plays a vital role in regenerative medicine. Regenerative medicine employs scientific and biological methods to repair functionally damaged tissues and organs that have been lost entirely or in part. Cell therapy, immunomodulatory therapy, and tissue engineering are all components of regenerative medicine. Patients who face clinical challenges such as end-stage organ failure and severe injuries would benefit from this method [113]. To determine the potential for regenerative medicine, adult stem cells (ASCs), iPSCs, and embryonic stem cells (ESCs) have all been extensively researched recently [114]. Until the advent of CR technology, it is challenging to effectively promote functional differentiation of stem cells. The CR approach can rapidly grow and expand cells with a more stem cell-like undifferentiated state and, more crucially, when CR cells are implanted in vivo, they can differentiate into lineage commitment, which is advantageous for conducting research in regenerative medicine. Using CR technology and tailored approaches, such as 3D scaffolds, cells can be cultured to create tissues that expertly recreate their original architecture and functionality and allow them to grow into functional tissues before implanting into the host. For instance, a research group engrafted CR airway epithelial cells and lung fibroblast onto a decellularized tracheal scaffold in a rabbit model, and after one week, demonstrated the presence of revascularization and keratin-positive cells across the scaffold [115]. Their findings indicate that this approach may enhance host epithelial healing and/or directly aid in mucosal regeneration, which could be valuable in regenerative medicine. Therefore, by utilizing patient-derived cells with stable genetic characteristics, CR technology will thereby address the unmet needs of tissue engineering for personalized regenerative medicine.

## 6. Limitations and Future Aspects

While CR holds immense potential for advancing cancer research and personalized medicine, it also has several limitations that warrant consideration. One significant limitation is the requirement for specialized culture conditions and expertise, which may limit its widespread adoption and scalability. Establishing and maintaining CR cultures demands technical proficiency and resources, including specific culture media formulations, equipment, and trained personnel [21,22]. This may pose challenges for research laboratories with limited infrastructure and funding, hindering the accessibility and reproducibility of CR-based studies.

Furthermore, CR technology faces challenges related to the representation of tumor heterogeneity and microenvironmental interactions. While CR cultures faithfully recapitulate the genomic and phenotypic characteristics of patient tumors, they may not fully capture the complex tumor microenvironment and stromal interactions that influence cancer progression and drug responses [116]. This limitation underscores the need for complementary models, such as PDX and organoids, to complement CR-based studies and provide a more comprehensive understanding of cancer biology. For example, the success of PDX and organoids will be largely increased when CR cultures are subjected to PDX and organoids.

Additionally, the applicability of CR in studying certain cancer types or patient populations may be limited. CR technology relies on the successful establishment of patient-derived cell cultures, which may be challenging for tumors with low cell viability or limited proliferative capacity. Moreover, the representativeness of CR cultures may vary across different cancer types and patient cohorts, potentially limiting the generalizability of findings [40].

Despite these limitations, ongoing research efforts aimed at optimizing CR protocols, addressing technical challenges, and integrating CR with complementary model systems offer opportunities to overcome current limitations. By leveraging CR technology in conjunction with other cutting-edge approaches, researchers can continue to advance our understanding of cancer biology, drug discovery, and personalized medicine, ultimately improving outcomes for patients with cancer.

## 7. Conclusions

The development of CR technology in the investigation of PCa opens fascinating research possibilities. CR approach can effectively and robustly establish cell lines from normal and tumor tissues. Once CR conditions are eliminated, CRCs can restore the lineage commitment of cells and maintain the developmental potential of the original tissue. CR cells can also be established from xenografts and organoid tissues. As a result, CR technology may be a good in vitro model to study different types of PCa from initiation to malignancy, allowing for the development of precision medicine and novel drug discovery. Future CR research may support tissue engineering for individualized regenerative medicine and offer the intriguing prospect of developing a living biobank for a wide biological spectrum of PCa.

## Figures and Tables

**Figure 1 cells-13-01005-f001:**
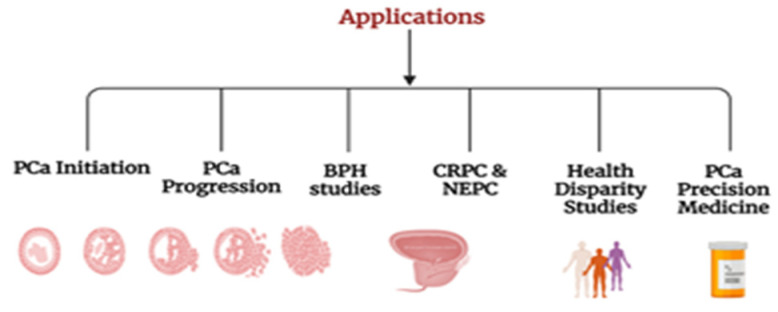
Application of CR technology to study PCa.

**Figure 2 cells-13-01005-f002:**
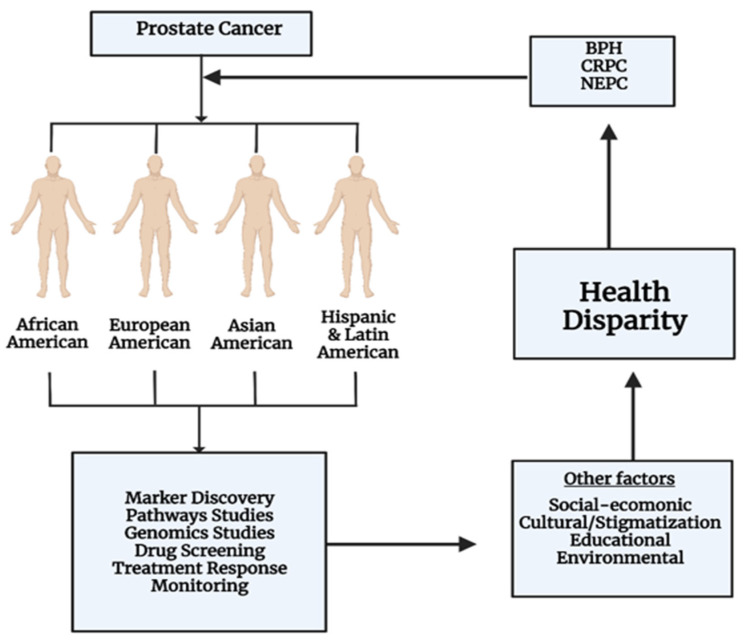
Understanding of the health disparity in study of PCa. The diagram delineates the multifaceted contributors to health disparities in PCa, focusing on ethnic groups such as African American, European American, Asian American, and Hispanic and Latin American. It features critical research domains like marker discovery, pathways studies, and genomics studies, which are pivotal in both understanding and influencing these disparities. Beyond genetic predispositions, the diagram acknowledges that disparities are further exacerbated by social-economic, cultural, educational, and environmental factors. Additionally, it recognizes the inherent diversity of PCa presentations, with conditions like BPH, CRPC, and NEPC being indicative of the disease’s variable impact on health outcomes.

**Figure 3 cells-13-01005-f003:**
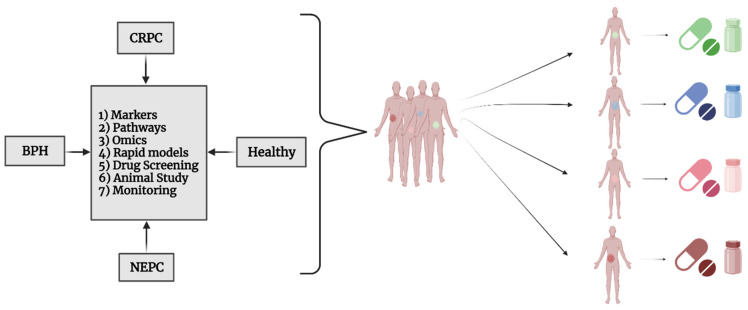
Personalized marker identification for precision medicine for the PCa. This figure illustrates the spectrum of PCa conditions—including BPH (benign prostatic hyperplasia), CRPC (castration-resistant PCa), and NEPC (neuroendocrine PCa)—and highlights the role of health disparities in the variation of disease states from one individual to another. It underscores the importance of personalized medicine research in identifying unique markers for each patient, with the goal of providing tailored treatments. The central box outlines the components of this research approach: (1) markers, (2) pathways, (3) omics (encompassing genomics, proteomics, and other related fields), (4) rapid models, (5) drug Screening, (6) animal Study, and (7) monitoring.

**Table 1 cells-13-01005-t001:** Comparison of different cancer models for the study of PCa. This table provides a comparison of various cancer models commonly used in PCa research. Each row represents a different type of sample origin or characteristic, while the columns represent specific parameters used to evaluate the effectiveness and suitability of each model. The parameters include sample origin, timing, success rate, rapid expansion, genetic stability, cost, life span, difficulty of differentiation, biobanking feasibility, tissue-specific characteristics, genetic manipulation capabilities, tumor–stromal interaction, and representation of primary tissue. Types of biopsy procedures: fine-needle aspiration (FNA), core biopsy, surgical specimens, and urine-derived cells. Types of tissues from above procedures: cancerous tissue, noncancerous tissue (for example, adjacent normal tissue). Cryopreserved tissue: frozen tissue samples using cryopreserved medium, including both cancerous and noncancerous prostate tissue, obtained from various sources like biopsies or surgical resections. −: not applicable. −/+: not applicable or impossible in general, but very few reports indicated very low level or minimal presence of the attribute. +: low level or minimal presence of the attribute. ++: moderate level or presence of the attribute. +++: high level or substantial presence of the attribute.

Sample Origin	Conventional Cell Lines	Primary Cells	PDX Model	3D Organoid	CR Cells
FNA	−	−	−	−/+	+++
Core biopsy	−	+	−	+	+++
Surgical specimens	+	++	++	+++	+++
Cryopreserved tissue	−/+	+/++	−/+	+++	+++
Cancerous tissue	+++	++	++	+++	+++
Noncancerous tissue	−	−/+	−	+	+++
Urine derived cells	−	−	−	−	+
Timing	Several days	1 to 4 weeks	1 to 5 months	1 to 4 weeks	1 to 10 days
Success rate	+	++	++	++	+++
Rapid expansion	+++	++	+	++	+++
Genetic Stability	+	++	++	++	++
Cost	+	++	+++	++	+
Life span	+++	+	+	++	+++
Difficulty of differentiation	+++	+	+++	+	+
Biobanking	−	+	++	+++	+++
Tissue-specific	+	+++	+++	+++	+++
Genetic manipulation	+++	−/+	−	++	++
Tumor–stromal interaction	−	−	++	+	−
Representation of primary tissue	+	++	++	++	++

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
