# Peer review of "Patient-Derived Conditionally Reprogrammed Cells in Prostate Cancer Research"

_cells, 2024, doi:10.3390/cells13121005_

Round 1

Reviewer 1 Report (New Reviewer)

Comments and Suggestions for Authors This is a comprehensive review of the applications of conditional reprogramming to grow prostate cancer cell. The review article is well structured and well written.

I only have minor suggestions:
1) While the review article goes in depth into the applications of conditional reprogramming in prostate cancer, those not familiar with it may benefit from a more in-depth explanation of the technique itself, and the technical nuances and considerations. Even a paragraph or a schematic may be adequate. 2) Line 115, topic "1. Patient-Derived Primary Cell Cultures in Prostate Cancer Research": The manuscript may benefit from a few lines about the molecular machinery and common pathways involved in prostate cancer development. 3) For the topic of using CRC in early clinical trials, may there be any concerns regarding the standardization of the technique across different labs and different countries? 4) Line 260–278 seems a bit wordy and redundant, it may benefit from consolidation. 5) Line 401, please correct "indomethacnin". 6) Some abbreviations are introduced more than once in the manuscript, for example in line 420 and 549 and 624 "patient-derived xenograft (PDX)" is no more necessary. 7) Line 581, referring to Dr. Shvets, is there a publication or reference for this that can be used instead of the individual's last name only? 8) Lines 35 and 573 until the end of the paragraph seem to have a different font or size.    

Author Response

This is a comprehensive review of the applications of conditional reprogramming to grow prostate cancer cell. The review article is well structured and well written. I only have minor suggestions: 1) While the review article goes in depth into the applications of conditional reprogramming in prostate cancer, those not familiar with it may benefit from a more in-depth explanation of the technique itself, and the technical nuances and considerations. Even a paragraph or a schematic may be adequate.

RE: Thanks for your suggestion. We added two paragraphs discussing the technique and culture media under ‘Role of Conditional Reprogramming (CR) in Cell Culture’

2) Line 115, topic "1. Patient-Derived Primary Cell Cultures in Prostate Cancer Research": The manuscript may benefit from a few lines about the molecular machinery and common pathways involved in prostate cancer development. 

RE: Thanks for your comments. We added a paragraph discussing the molecular machinery and common pathways involved in prostate cancer development.

3) For the topic of using CRC in early clinical trials, may there be any concerns regarding the standardization of the technique across different labs and different countries?

RE: The protocol for conditional reprogramming (CR) cell culture technique appears to be well-established, as evidenced by multiple studies and resources discussing its mechanisms, applications, and development. However, CR, relies heavily on well-trained personnel. Proper training equips laboratory staff with the necessary skills and knowledge to execute protocols accurately, minimizing errors and ensuring consistency.

4) Line 260–278 seems a bit wordy and redundant, it may benefit from consolidation

RE: Thanks. We consolidated the lines 260- 278.

5) Line 401, please correct "indomethacnin". 

RE: Thanks. it was corrected.

6) Some abbreviations are introduced more than once in the manuscript, for example in line 420 and 549 and 624 "patient-derived xenograft (PDX)" is no more necessary.

RE: Thanks. it was corrected.

7) Line 581, referring to Dr. Shvets, is there a publication or reference for this that can be used instead of the individual's last name only?

RE: Thanks. It was corrected.

8) Lines 35 and 573 until the end of the paragraph seem to have a different font or size.    

RE: Thanks. We corrected.

Reviewer 2 Report (New Reviewer)

Comments and Suggestions for Authors

This review focuses on the use of conditional reprogrammed patient-derived primary cultures of prostate cancer cells.  This technique is being used more and more frequently and has several advantages, as well as limitations. While the review is well intentioned, it is poorly written.  It’s very redundant and repeats the same message each section.  It’s clear that the authors are pushing conditional reprogramming and the review is somewhat bias in its writing.  Further, the authors assume that the reader knows what conditional reprograming is and how it’s done.  No discussion of the history, methods and approach of conditional reprograming is given.  A limitation section is given, but only at the end and it seems cursory.  The article also changes tenses and viewpoints several times, appearing to be written by several authors.  These, and other concerns are listed below.

1.     The authors should define conditional reprograming as earlier as possible in the manuscript and briefly discuss how it different from other approaches, the methods used and its pro’s and con’s.

2.     The review is very redundant.  The same message is repeated at each section.  There is little to no flow, and it reads more like a propaganda piece for  conditional reprograming as opposed to an unbiased literature review.

3.     There are several one sentence paragraphs that simply state facts but are not put in context.

4.     Figure 1 is very low quality and not very useful.

5.     FNA needs to be defined in Table 1, as does the scoring system (what does each + represent)?

6.     The “P” in “progression” should be capitalized on page 7 section C.

7.     The word conditional before CR should be removed form line 317.

8.     Figures 2 and 3 do not really make any sense and is difficult to follow without figure legends.  They are not really the informative.

9.     It appears that there are several areas where the font is larger or of a different type.  For example, page 14, lines 573 to 586.  This supports the idea that this was written by several authors and time was not devoted to integrate their message.

10. A reference is needed for the development of the RICHI Assay (page 14 lines 582).  Furthermore, this appears to be a plug for the authors method.  I am not sure this is appropriate without an un-biased discussion of its advantages and limitations.

Author Response

  1. The authors should define conditional reprograming as earlier as possible in the manuscript and briefly discuss how it different from other approaches, the methods used and its pro’s and con’s.

RE: Thanks for your comments. We added two paragraphs at the end of introduction,

  1. The review is very redundant.  The same message is repeated at each section.  There is little to no flow, and it reads more like a propaganda piece for  conditional reprograming as opposed to an unbiased literature review.

RE:  Thank you for your constructive feedback. The perception of bias in the writing may stem from our intimate involvement in the development of the CR protocol, which has afforded us a deep understanding of its significance and applications. However, we have included comprehensive comparisons between our CR protocols and other existing methodologies.

We have revised and eliminate redundancies from [introduction, C. Acquisition of Stem Cell Properties, 3. Advantages of CR over Traditional Approaches, A. Patient-Derived Models for Precision Medicine in Prostate Cancer] and refined Section B (Limitations of Traditional Prostate Cancer Cell Lines) to include a more detailed analysis of traditional cell cultures in prostate cancer (PCa), providing specific examples to enhance clarity and scholarly rigor. In addition, we added a section discussing the CR technique and culture media under ‘Role of Conditional Reprogramming (CR) in Cell Culture’ and a paragraph on Optimization of CR. And included more detailed explanation of CR and its optimization conditions. We acknowledge the novelty of our model (Conditional Reprogramming) in PCa research, which explains the current scarcity of direct examples in the literature. To address this, we have made efforts to discuss the potential applications of CR in a balanced manner, ensuring that our review is comprehensive and objective. We have also incorporated comparative analyses where possible to underscore the advantages of CR within the context of existing research paradigms. We believe these changes improve the flow of the manuscript and present CR as an emerging method within the broader context of PCa research.

  1. There are several one sentence paragraphs that simply state facts but are not put in context.

RE:  We refined and eliminated one sentence paragraphs from (1. Patient-Derived Primary Cell Cultures in Prostate Cancer Research, B. Limitations of Traditional Prostate Cancer Cell Lines). We remain open to more specific reviewal suggestions and are willing to make additional amendments to particular paragraphs.

  1. Figure 1 is very low quality and not very useful.

RE:  The intention of Figure 1 is to provide an overview of the diverse applications of CR, which will be discussed in detail in subsequent paragraphs. Its purpose is to orient the reader and preview the content that follows in the manuscript.

  1. FNA needs to be defined in Table 1, as does the scoring system (what does each + represent)?

RE:  Thanks. We added table legend:  This table provides a comparison of various cancer models commonly used in prostate cancer research. Each row represents a different type of sample origin or characteristic, while the columns represent specific parameters used to evaluate the effectiveness and suitability of each model. The parameters include sample origin, timing, success rate, rapid expansion, genetic stability, cost, life span, difficulty of differentiation, biobanking feasibility, tissue-specific characteristics, genetic manipulation capabilities, tumor-stromal interaction, and representation of primary tissue. Different types of samples, such as fine-needle aspiration (FNA), core biopsy, surgical specimens, cryopreserved tissue, cancerous tissue, noncancerous tissue, and urine-derived cells. +: Low level or minimal presence of the attribute. ++: Moderate level or presence of the attribute. +++: High level or substantial presence of the attribute.

  1. The “P” in “progression” should be capitalized on page 7 section C.

RE:  Thank you for the notice; it has been corrected.

  1. The word conditional before CR should be removed form line 317.

RE: Thank you for the notice; it has been corrected.

  1. Figures 2 and 3 do not really make any sense and is difficult to follow without figure legends.  They are not really the informative.

RE:  We added figure legends, explaining each figure.

  1. It appears that there are several areas where the font is larger or of a different type.  For example, page 14, lines 573 to 586.  This supports the idea that this was written by several authors and time was not devoted to integrate their message.

RE:  Thank you for the notice; it has been corrected.

  1. A reference is needed for the development of the RICHI Assay (page 14 lines 582).  Furthermore, this appears to be a plug for the authors method.  I am not sure this is appropriate without an un-biased discussion of its advantages and limitations.

RE:  Thank you for your input. We added a reference.

Round 2

Reviewer 2 Report (New Reviewer)

Comments and Suggestions for Authors

The manuscript is improved, but it's still not what I consider a quality review.  The figures are basic and the data don't really add to the literature extensively.  

Comments on the Quality of English Language

Fine

Author Response

I decided not to address the comments from reviewer 2 since the comments are not specific, and unprofessional…… To understand his/her “what I consider … ” is an impossible task for any scientist.

This manuscript is a resubmission of an earlier submission. The following is a list of the peer review reports and author responses from that submission.

Round 1

Reviewer 1 Report

Comments and Suggestions for Authors

The content of this manuscript is far from its title. The authors have written a long introduction on prostate cancer, epidemiology, molecular phenomena, treatments, etc., and the least discussed topics are experimental models, such as reprogrammed cells. The reader looking for information on modern experimental models to study this disease will not find it in this article, despite its title.

Reviewer 2 Report

Comments and Suggestions for Authors

To the reviewer:

Thank you for your review about applications of CR cells in prostate cancer, especially in initiation and progression of prostate cancer and its molecular pathogenesis, health disparity, and the role in development of personalized medicines

The topic of CR is viewed with a lot of effort and >150 citations. The review looks like a non-exhaustive list of information. Unfortunately, the main thread of “conditional reprogramming” is very small compared to the side stages. Finally, author's purpose and the objective of the article seem unclear.

General improvements:

-       Please focus on the main topic of “conditional reprogramming”

-       Please slash secondary information and reduce number of citations

-       Please enhance the scientific benefit of your work

-       Please introduce a leitmotiv

Special improvements:

-       Introduce abbreviations correctly e.g. PTEN, MYC, BCRA1, FAK...

-       Please describe Table 1, there is no legend. 

-       Please describe the CRC method more in detail and compare it more detailed with other in vitro models. 

-       You have described PCa initiation and progression in great detail, what is the significance of CR here? Which data comes from CRC studies? Please specify.

-       The paragraph “CRPC AND NEPC” is very general about treatment options and markers. Please specify the role of CR and which data are already published.

Best regards